# Guided Policy Search for Parameterized Skills using Adverbs

**Benjamin A. Spiegel** [1]   **George Konidaris** [1]

## Abstract

We present a method for using adverb phrases to adjust skill parameters via learned *adverb-skill groundings*. These groundings allow an agent to use adverb feedback provided by a human to directly update a skill policy, in a manner similar to traditional local policy search methods. We show that our method can be used as a drop-in replacement for these policy search methods when dense reward from the environment is not available but human language feedback is. We demonstrate improved sample efficiency over modern policy search methods in two experiments.

## 1. Introduction

In collaborative human-robot environments, embodied agents must be capable of integrating language commands into behavior. Natural language instruction can range from feedback on the subtlest of movements to abstract, high-level plans. While humans are generally capable of giving and receiving language feedback regarding all aspects of a task, methods for integrating language understanding into behavior differ based on the level of behavioral abstraction the instruction refers to. Following (Rodriguez-Sanchez et al., 2020), who argued that the structure of language closely relates to the structure of an agent's decision process, we focus on grounding *adverbs*—words used to describe the quality of a verb (i.e. a skill)—to directly modify skill execution.

We adopt the framework of hierarchical reinforcement learning (Barto & Mahadevan, 2003), wherein an agent's behavior is mainly generated by skills responsible for low-level motor control, and learning is primarily concerned with sequencing given skills to solve a task. Much existing research on integrating language understanding into hierarchical agents attempts to map language to sequences of abstract skill executions (Andreas et al., 2017; Mei et al., 2016; Oh et al., 2017). However, agents must also be able to use language to modify their underlying skill policies. Commands like "lift the pallet *higher*" and "crack the egg *gently*" clearly request adjustments to a specific skill execution. Therefore, a key question is how natural language understanding can ground to changes in the lowest levels of behavior.

The existence of adverbs that modify discrete verbs calls for agents with a discrete set of skills, with behavior that can be modified by a parameter vector describing how the skill can be executed (Da Silva et al., 2012; Masson et al., 2016). We propose a novel method for grounding adverbs to adjustments in skill parameters—called *adverb-skill groundings*—which, when integrated into policy search, lead to greater sample-efficiency than traditional policy search methods that typically depend on explicit reward from the environment. We demonstrate the effectiveness of adverb-skill groundings for policy search in a toy ball-throwing domain and a domain involving a simulated 7-DoF robot arm. We compare the sample efficiency of our approach to $PI^2$-CMA (Stulp & Sigaud, 2012), a state-of-the-art local policy search method.

## 2. Background

### 2.1. Mechanisms of Hierarchical Control

Reinforcement learning tasks are typically modeled as Markov decision processes (MDPs). An MDP can be represented by the tuple $\langle S, A, R, T, \gamma \rangle$, where $S$ is the set of states, $A$ is the set of actions, $R$ is the reward function, $T$ is the transition function, and $\gamma$ is the discount factor. The goal of an agent is to find a policy, $\pi(a|s)$—a function that selects an action for each state—which maximizes the expected sum of discounted rewards:

$$\mathbb{E}_\pi \left[ \sum_{t=0}^{\infty} \gamma^t R(s_t, a_t, s_{t+1}) \right].$$

Masson et al. (2016) introduced Parameterized Action Markov Decision Processes (PAMDPs) to model situations where agents have access to discrete actions that are parameterized by real-valued vectors. For each step in a PAMDP, an agent must choose a discrete action $a \in A_d$ and a cor-

[1]Department of Computer Science, Brown University, Providence, RI. Correspondence to: Benjamin A. Spiegel <bspiegel@cs.brown.edu>.

*Interactive Learning with Implicit Human Feedback Workshop at ICML 2023*, Honolulu, Hawaii, USA. PMLR 202, 2023. Copyright 2023 by the author(s).

responding continuous parameter $x \in X_a \subseteq \mathbb{R}^{m_a}$. As pointed out by Rodriguez-Sanchez et al. (2020), parameterized actions are a plausible target for grounding adverbs, because adverbs modify verbs in a similar fashion to how continuous parameters modify actions in a PAMDP. This work refines and formalizes this relationship by grounding adverbs directly to adjustments in skill parameters.

## 2.2. Policy Search Methods

Policy gradient methods (Sutton et al., 1999) are a collection of local search methods which optimize a policy by leveraging the gradient of a performance metric. Formally, they optimize a parameterized policy $\pi_\theta$ based on the gradient of a scalar performance metric $J(\theta)$, using an update rule similar to gradient ascent:

$$\theta_{t+1} = \theta_t + \alpha \widehat{\nabla J(\theta_t)}.$$

One common strategy which is shared across multiple methods for local policy search involves leveraging weighted probability averaging to perform parameter updates as opposed to using estimations of the gradient of the performance metric, which can be noisy (Theodorou et al., 2010). The structure of these algorithms is as follows: *1)* Sample $K$ parameters from a distribution; *2)* Sort the samples with respect to their performance given by $J$; *3)* Recalculate the distribution parameters based on the top $K_e$ 'elite' samples in the sorted list; *4)* Repeat this process with the newly calculated distribution until convergence or after a number of iterations. Algorithms within this class differ significantly in their implementation of these steps. For example, Cross Entropy Methods (CEM) (Mannor et al., 2003) recalculate the mean and covariance of the sampling distribution after each step, while Policy Improvement with Path Integrals (PI$^2$) (Theodorou et al., 2010) only updates the mean. In their design of PI$^2$-CMA, Stulp & Sigaud (2012) integrate a CEM-like probability-weighted averaging to update the covariance of the sampling distribution into the base PI$^2$ algorithm, enabling it to autonomously adapt its exploration.[1]

## 2.3. Representations of Natural Language

A crucial step in processing natural language for computation is encoding sequences of natural language tokens into mathematical representations that are semantically meaningful.[2] Most modern natural language processing research on semantic representation models units of language as high-dimensional vectors which are learned in an unsupervised fashion from large corpora (Mikolov et al., 2013; Pennington et al., 2014). These semantic space models leverage

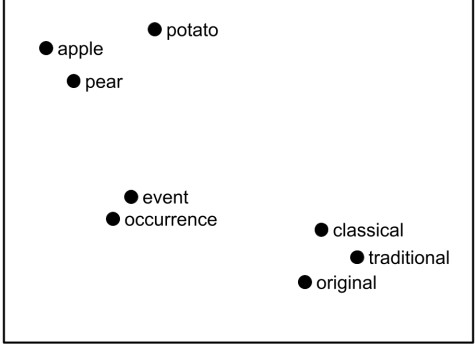

*Figure 1.* In this illustration of a semantic space, words are encoded as points in a vector space, and semantically similar words are located closely together. Semantic spaces which can capture the expressive power of language are often high-dimensional, representing features of language across many latent axes. Some semantic space representations can encode more complex units of language such as phrases or entire sentences. This illustration mirrors a figure in Mitchell & Lapata (2010).

the principle of *distributional semantics*, a hypothesis that semantic representations of words can be derived from the patterns of their lexical co-occurrences.[3] Semantic space models have the advantage of capturing language similarity as a geometric property—the distance between two language vectors or the cosine of the angle between them can be used as a measure of semantic similarity. Words are placed into a learned semantic space via an embedding function:

$$E : P \to \mathbb{R}^N,$$

mapping units of language $p$ to vectors in $\mathbb{R}^N$. A simplified example of a semantic space is shown in Figure 1.

## 3. Grounding Adverbs to Adjustments of Skill Parameters

We assume a language-conditioned, human-robot environment in which a human desires to use natural language to modify the behavior of an agent engaged in solving a task. In line with our goal to ground language to the lowest levels of behavior, we assume that the agent is equipped with a low-level motor skill $\Theta$, which induces a parameterized policy $\pi_{\Theta(\tau)} = \pi(a|s, \Theta(\tau))$ where $\tau \in T$. The goal of the agent is to find the desired action sequence it should take (corresponding to a hidden desired skill parameter $\tau^*$) in response to natural language criticism.

Interaction in the environment begins with the agent executing its motor skill using a random initial skill parameter $\tau_0$. The human and agent then engage in an iterative pro-

---

[1]See Stulp & Sigaud (2012) for a more thorough analysis of this class of direct policy improvement algorithms.

[2]See Markman (1998) for a more thorough account of semantic representations.

[3]This principle was famously summarized by Firth (1957) as, "you shall know a word by the company it keeps."

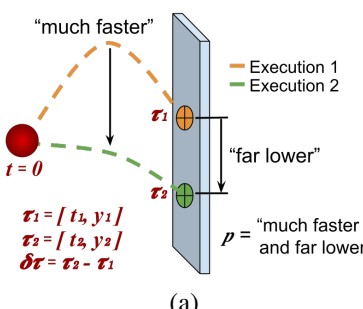
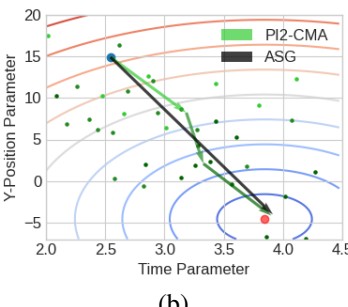
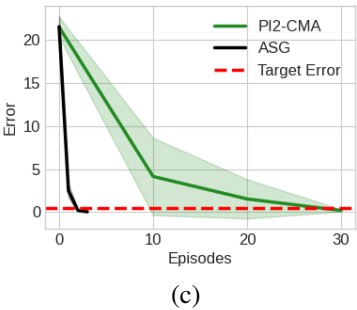

(a)             (b)             (c)

*Figure 2.* (a) Illustration of the Ball-Throw task. (b) Visualization of a sample policy search for the Ball-Throw task using PI$^2$-CMA and ASG plotted over error contours. The samples that PI$^2$-CMA collected are plotted in green. PI$^2$-CMA required 30 episodes to converge, while ASG required a single episode using an adverb phrase. (c) Error curves for the Ball-Throw task. Both policy searches terminate once error is below a certain threshold (dashed red line).

cess of behavior modification in which the human provides criticism regarding the agent's most recent skill execution in the form of an adverb phrase $l$, and the agent responds with an updated skill execution using a new skill parameter $\tau_{t+1}$. This process repeats until the human is satisfied with the skill execution, i.e. when $\tau_t$ is within $\epsilon$ of $\tau^*$.

The novel intuition presented in this work is that adverb phrases encode rich information about the agent's low-level skill performance. Specifically, they encode a desired adjustment, $\delta\tau$, to the policy parameters for a parameterized skill. To extract skill adjustments from adverb feedback, we propose an *adverb-skill grounding* (ASG), $\Lambda$, that maps adverb phrases, $l$, and skill parameters, $\tau$, to adjustments in skill parameters:

$$\Lambda : L \times T \to \mathbb{R}^{|T|}.$$

Using an ASG, an agent can perform a policy search without direct interaction with a reward function by iteratively adjusting its skill parameter using the following update rule:

$$\tau_{t+1} = \tau_t + \Lambda(l, \tau_t),$$

until the skill execution is satisfactory to the human. Since an ASG need only be learned once for a given skill, we demonstrate that they exhibit higher sample efficiency than traditional policy search methods which utilize episodic rewards.

Our objective is to learn $\Lambda$ for a skill $\Theta$ from a set $K$ of tuples $(l, \tau, \delta\tau)$—where $l$ is a vector embedding of an adverb phrase describing the behavior difference between the policies $\pi_{\Theta(\tau)}$ and $\pi_{\Theta(\tau+\delta\tau)}$—which can be done using a nonlinear regression model $\Phi$, mapping $(\tau, l) \to \delta\tau$. We can generate $K$ using the following procedure: *1)* Collect $|K|$ pairs of skill parameters $(\tau, \tau')$ sampled uniformly at random from $T$; *2)* For all pairs, execute the parameterized skill using each parameter, resulting in policies $(\pi_{\Theta(\tau)}, \pi_{\Theta(\tau')})$; *3)* Label the apparent difference between the policies with

an adverb phrase $l$. A human can do this after witnessing rollouts of the policies generated in the previous step. *4)* Calculate the difference between the skill parameters to generate a label: $\tau' - \tau = \delta\tau$.

### 3.1. Adverb Embeddings

Our model requires a method for grounding adverb phrases into meaningful vector embeddings. Modern semantic embedding methods are designed to capture any possible natural language fragment for a given language, and as a result they often represent language as vectors with many hundreds of latent axes. The specific use-case of language embeddings for our model requires them to capture a narrow class of linguistic items: adverbs of motion. For this reason, we designed a natural and intuitive embedding procedure that produces clear and concise vector embeddings from language. The dimensionality of our language embeddings match the number of unique *adverb axes* which are applicable to the specified task. For a task with a jumping agent, the adverb axes might be *higher-lower* and *right-left*. Modifiers like "a little" and "much" act as scalar multipliers to the adverbs they modify. For example, "A little higher and much more to the left" will be embedded as $[1, -3]$. Each axis is then normalized to between $-1$ and $1$. While these embeddings can easily be calculated with formal linguistic methods using a direct compositional approach (Jacobson, 2014), future work should relax this constraint.

## 4. Experiments

We evaluated our model using two tasks that are prime targets for a parameterized skill. The first is a simple ball-launching task in which the goal of the agent is to throw a ball to reach a target at a specified location and time. The second task uses a simulated 7-DoF MuJoCo (Todorov et al., 2012) robot in a modified OpenAI Gym (Brockman et al., 2016) Fetch Slide environment, where the agent's goal is to

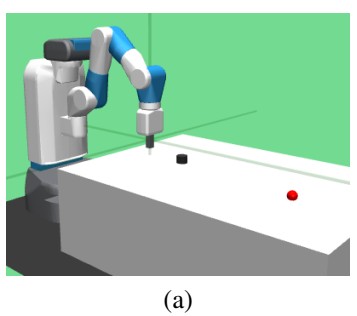
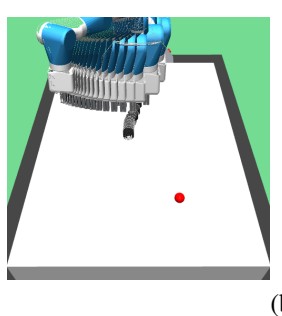
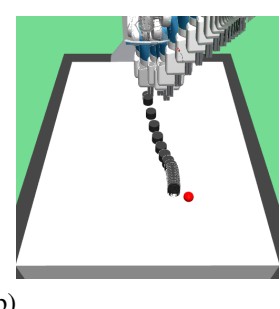
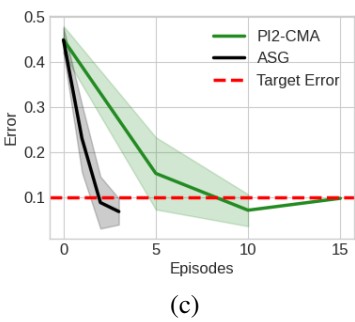

(a)              (b)              (c)

*Figure 3.* (a) The Fetch Slide task in which the goal is to hit a puck to a specified position (in red). (b) Visualization of a single episode of an ASG policy search which uses the adverb phrase, "much lower and to the right." (c) Error curves for the Fetch Slide task. Both policy searches terminate once error is below a certain threshold (dashed red line).

hit a puck to a randomized target location.[4] We compare our simple policy update procedure using ASG to $PI^2$-CMA, a direct policy search algorithm that optimizes over the space of skill parameters.

### 4.1. Ball-Throw Task

In the Ball-Throw task, an agent must throw a ball at a target located on a wall, $y_{goal}$, at a specific time, $t_{goal}$, where $\tau = [t_{goal}, y_{goal}]$. The policy of the agent $\pi_\theta$ is parameterized by a vector $\theta \in \mathbb{R}^2$ which stores the vertical and horizontal components of velocity applied to the ball at the origin at time $t_0$. We implemented a parameterized skill $\Theta : T \rightarrow \mathbb{R}^2$—converting task parameters to policy parameters—using Newtonian physics:

$$\Theta(\tau) = [\frac{1}{t_{goal}}, 5t_{goal} + \frac{y_{goal}}{t_{goal}}].$$

While adverb-skill groundings do not require environmental reward, $PI^2$-CMA does, so we define a dense reward function as the $l^2$-norm of the difference between the target and current task parameters: $R(\tau) = -|\tau - \tau^*|_2$.

The adverbs most relevant to this task describe the target speed and location of the ball, so we designed a simple 2-dimensional adverb embedding to capture two primitive adverb axes: *faster-slower* and *higher-lower*. We collected only 30 samples of training data in the form $(l, \tau, \delta\tau)$ using the procedure defined in section 3, which we labeled with adverbs using the procedure in Algorithm 1.

We chose a feed-forward neural network as our regression model $\Phi$ for learning the adverb-skill grounding. As shown in Figure 2(c), adverb-skill groundings converged to within the target error bound using 32 less episodes on average than $PI^2$-CMA, which required 10 reward sampling episodes for every policy update. Across 100 trials, a single episode

using an ASG achieved the same decrease in error as approximately 16.4 episodes of $PI^2$-CMA.

### 4.2. Fetch Slide Task on a Simulated Robot

We modeled a "fetch" policy on a the simulated 7-DoF robot arm using Dynamic Movement Primitives (DMPs) (Schaal et al., 2003) due to their successful application in robotics. We learned a parameterized skill using a K-Nearest Neighbors regression model—mapping puck goal positions, $\tau \in \mathbb{R}^2$, to DMP parameters, $\theta \in \mathbb{R}^{13}$—using skill executions generated using $PI^2$-CMA (Stulp & Sigaud, 2012) as implemented by Abbatematteo et al. (2021).

The adverb axes that were most well-suited for describing this task were *higher-lower* and *left-right*. We collected 50 samples of training data in the form $(l, \tau, \delta\tau)$ using the procedure described in 3. We labeled the data using Surge AI,[5] a crowd-sourced data-collection service that uses human workers. Workers were shown two images of a skill execution (similar to the images in Figure 3 (b)) and asked to rate them by how much the puck in the second image was higher, lower, to the left, and to the right of the puck in the first image. Their responses were then embedded using the procedure defined in 4. The environment came with a dense reward function, which was only used in the $PI^2$-CMA runs.

We chose a multi-output Support Vector Machine (Vapnik, 2000) as our regression model $\Phi$ for learning the adverb-skill grounding. As shown in Figure 3(c), adverb-skill groundings converged to within the target error bound using 12 less episodes on average than $PI^2$-CMA, which required 5 reward sampling episodes for every policy update. Across 18 trials, a single episode using an ASG achieved the same decrease in error as approximately 4.6 episodes of $PI^2$-CMA.

---

[4]Code for both experiments is available on GitHub: (https://github.com/SkittlePox/thesis-lang-skill-params).

[5]Access to Surge AI via https://www.surgehq.ai/.

**Algorithm 1** Adverb Labeling and Embedding Procedure (Ball-Throw)

---

**Input**: Skill Parameters, $(\tau, \tau')$
**Output**: Adverb Embedding, $l$

$\quad \delta\tau \leftarrow \tau' - \tau$
$\quad t_{adv} \leftarrow 0$
$\quad y_{adv} \leftarrow 0$
$\quad$**if** $|\delta\tau_0| > 0.05 + 0.15(-\tau_0 + 4)$ **then**
$\quad\quad$**if** $|\delta\tau_0| > 1.2 + 0.17(-\tau_0 + 4)$ **then**
$\quad\quad\quad t_{adv} \leftarrow 3$ {"much"}
$\quad\quad$**else if** $|\delta\tau_0| > 0.6 + 0.15(-\tau_0 + 4)$ **then**
$\quad\quad\quad t_{adv} \leftarrow 2$
$\quad\quad$**else**
$\quad\quad\quad t_{adv} \leftarrow 1$ {"a little"}
$\quad\quad$**end if**
$\quad$**end if**
$\quad$**if** $|\delta\tau_1| > 0.5 + 0.15(\tau_1 + 15)$ **then**
$\quad\quad$**if** $|\delta\tau_1| > 12 + 0.17(\tau_1 + 15)$ **then**
$\quad\quad\quad y_{adv} \leftarrow 3$ {"much"}
$\quad\quad$**else if** $|\delta\tau_0| > 0.6 + 0.15(-\tau_0 + 4)$ **then**
$\quad\quad\quad y_{adv} \leftarrow 2$
$\quad\quad$**else**
$\quad\quad\quad y_{adv} \leftarrow 1$ {"a little"}
$\quad\quad$**end if**
$\quad$**end if**
$\quad$**if** $\delta\tau_0 \geq 0.0$ **then**
$\quad\quad y_{adv} \leftarrow -y_{adv}$ {"slower," not "faster"}
$\quad$**end if**
$\quad$**if** $\delta\tau_1 \leq 0.0$ **then**
$\quad\quad y_{adv} \leftarrow -y_{adv}$ {"higher," not "lower"}
$\quad$**end if**
$\quad$**return** $[t_{adv}, y_{adv}]$

---

| | Episodes to Converge | |
| --- | --- | --- |
| | Ball-Throw | Fetch Slide |
| PI$^2$-CMA | 34.1 | 16.3 |
| ASG | 2.1 | 3.6 |

*Table 1.* Results of Ball-Throw and Fetch Slide experiments. We ran 100 trials of the Ball-Throw task and 18 trials of the Fetch Slide task. An adverb was worth approximately 16.4 reward samples in the Ball-Throw task and 4.6 reward samples in the Fetch Slide task. Fast convergence occurs because the policy search is performed over low-dimensional action parameter space.

# 5. Related Work

## 5.1. Natural Language in Reinforcement Learning

Most existing research that has used natural language in reinforcement learning problems can be categorized as either *language-conditional* (in which agents must interact with language to solve problems) or *language-assisted* (in which language can be used to facilitate learning) (Luketina et al., 2019). Our setting is language-conditional, since the agent is presumed only to have access to natural language feedback. Though, prior works across both categories are relevant.

Some previous research has attempted to map language instructions to reward functions. Arumugam et al. (2017) map natural language instructions to goal-state reward functions at multiple levels of abstraction within a planning hierarchy over an object oriented MDP formalism. While this approach has the advantage of being able to interpret instructions at multiple levels of abstraction, the base-level actions used in the approach are significantly more abstract than motor skills. Goyal et al. (2019) shape reward functions by interpreting previous actions to see if they can be described by a natural language instruction, effectively grounding natural language directly to action sequences.

In contrast, our work grounds atomic language fragments directly to continuous skill parameters, which allows us to make granular adjustments to the execution of motor skills via language commands.

Other research has mapped symbolic instructions directly to policy structure. Andreas et al. (2017) learn a mapping from symbolic policy sketches to sequences of modular sub-policies in the form of options (Sutton et al., 1999). Shu et al. (2017) use language instructions to help a hierarchical agent decide whether to use a previous skill or to learn a new one. Hu et al. (2019) similarly map language instructions to macro-actions in a real-time strategy game, which are then performed using a separate model. Tellex et al. (2011) generate high-level plans from the semantic structure of language instructions. Gopalan et al. (2020) derive symbol sketches from demonstrated navigation trajectories which they ground language instructions to. While all of these works ground language to agent behavior, ours is the first to integrate language feedback at the level of modifying the low-level behavior of motor skills.

## 5.2. Semantic Representation

While earlier semantic space representations were mainly concerned with encoding individual words or n-grams into vector space (Lund & Burgess, 1996; Landauer & Dumais, 1997), there has been recent discussion regarding how to capture phrases and sentences with similar machinery. Mitchell & Lapata (2010) explore this problem, which hinges on linguistic structures as being *compositional*, i.e. that the meaning of a language fragment is a function of the meanings of its composite parts. Compositionality itself has been accounted for in older logic-based formalisms (Montague, 1974), but incorporating compositionality into modern semantic space representations is still an unsolved problem.

Baroni & Zamparelli (2010) proposed a candidate solution that accounts for compositionality in semantic space models by representing nouns as vectors and adjectives as matrices, and the meaning of their combinations to be their tensor products. Krishnamurthy & Mitchell (2013) expand on

this idea by using Combinatory Categorial Grammar (CCG) (Steedman, 1996) to prescribe tensors of various modes to syntactic categories, whose weights they learn via a training process that utilizes a corpus.

While the primary focus of this research is not on semantic models, we firmly believe that core linguistic principles—such as the principle of compositionality—should be considered when designing systems for grounding language to behavior. Accordingly, we utilized the syntax/semantics formulation laid out by Steedman (1996) and the intuition behind more recent compositional distributional semantics research (Baroni & Zamparelli, 2010; Krishnamurthy & Mitchell, 2013) in our strategy for grounding adverbs. Adverbs by the CCG account are functions from verbs to verbs, and adverbs by our account are similarly functions from skills to skills.

## 6. Conclusion

We have presented a novel method for efficiently integrating granular natural language feedback into low-level behavior. The method relies on learning *adverb-skill groundings*, mappings of adverbs to adjustments in skill parameters, which can be learned once using few training examples and do not require reward from the environment. Using adverb-skill groundings, an agent can integrate adverb feedback into a policy search—in place of sample-based direct policy search methods—and achieve an order of magnitude increase in sample efficiency.

This work can be extended in several directions. First, the adverbs that are model is capable of using for skill modification are limited by the parameterization of the skill. If no variation in skill parameters could result in the desired effect of an adverb, a new skill parameterization should be learned with greater expressive power. Second, humans typically do not need to learn to ground adverbs each time they learn a new skill; once they understand an adverb, they can apply it to many skills. Future work might consider making technical accommodations to enable this.

Another important question to address is how language should be embedded for usage in modifying skill parameters. Our embedding procedure was designed by a human expert with knowledge of the skills and the adverbs that are most applicable to them. Future work should look to relax this constraint, perhaps by defining a broad and exhaustive set of adverbs of motion which can be embedded and applied to any skill.

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
