# OpenReview forum: "Guided Policy Search for Parameterized Skills using Adverbs"
_ICML.cc/2023/Workshop/ILHF — ILHF Workshop ICML 2023_

### Official Review · Reviewer_3M5c · 2023-06-11
**Review for Submission #1`**

**Rating:** 5
**Confidence:** 4

**Review:**

The paper uses adverb phrases from human language feedback to adjust skill parameters. The authors attempt to extract the information encoded in the adverb phrases to guide the adjustment of the the agent’s low-level skill. The authors also provide experiments on ball-throw and fetch slide to justify the effectiveness of the proposed method.

My main concern with this paper is the following:

1. In the proposed ASG algorithm, it is unlcear how one obtains the map from adverb phrases in natural language to adjustment. Why are 30-50 examples enough to learn the mapping in the experiments? What are the loss functions?

2. Where do the constants in Algorithm 1 come from? Do we need to define different mappings manually as in Algorithm 1 for all other tasks?

Overall, I think the proposed question and algorithm are interesting. But the paper needs a significant improvement in clarity and presentation.

---

### Official Review · Reviewer_yfLL · 2023-06-15

**Rating:** 7
**Confidence:** 3

**Review:**

I enjoyed reading this paper. It enables end-users to change an agent’s skill by using adverbs. The writing is clear overall and the approach seems novel.

As the authors mentioned in the conclusion, the approach as is does not seem very generalizable because the embeddings seem hand-designed. In future work, the authors could explore using pre-trained embeddings to be able to use a wider range of adverbs.

Overall, the paper is interesting and seems like a good fit for this workshop.

---

### Decision · Program_Chairs · 2023-06-20

Accept